# Inhibition of the Sodium–Calcium Exchanger Reverse Mode Activity Reduces Alcohol Consumption in Rats

**DOI:** 10.3390/ijms25074132

**Published:** 2024-04-08

**Authors:** Gleice Kelli Silva-Cardoso, Prosper N’Gouemo

**Affiliations:** Department of Physiology and Biophysics, Howard University College of Medicine, Washington, DC 20059, USA; gleicekelli.cardoso@howard.edu

**Keywords:** alcohol intake, alcohol preference, sodium–calcium exchanger, KB-R7943, SN-6, water intake

## Abstract

Excessive and uncontrolled consumption of alcohol can cause alcohol use disorder (AUD), but its pharmacological mechanisms are not fully understood. Inhibiting the reverse mode activity of the sodium–calcium exchanger (NCX) can reduce the risk of alcohol withdrawal seizures, suggesting that NCX could play a role in controlling alcohol consumption. Here, we investigated how two potent inhibitors of NCX reverse mode activity, SN-6 (NCX1) and KB-R7943 (NCX3), affect voluntary alcohol consumption in adult male and female rats using the intermittent alcohol access two-bottle choice paradigm. Initially, animals were trained to drink 7.5% ethanol and water for four weeks before administering SN-6 and KB-R7934. Afterward, their alcohol intake, preference, and water intake were recorded 2 and 24 h after exposure to water and 7.5% ethanol. SN-6 significantly reduced alcohol consumption by 48% in male and 36% in female rats without affecting their water intake. Additionally, SN-6 significantly reduced alcohol preference in females by 27%. However, KB-R7943 reduced alcohol consumption by 42% in female rats and did not affect alcohol preference or water intake. These findings suggest that alcohol exposure increased NCX reverse activity, and targeting NCX1 could be an effective strategy for reducing alcohol consumption in subjects susceptible to withdrawal seizures.

## 1. Introduction

Alcohol is a widely used substance around the world. Excessive and uncontrolled alcohol consumption can lead to the development of an alcohol use disorder (AUD), which is linked with higher rates of mortality and morbidity [1,2,3,4]. Abstaining from alcohol can alleviate AUD symptoms. However, it can also cause a condition known as alcohol withdrawal syndrome, which can result in alcohol withdrawal seizures and delirium tremens [5,6]. Although pharmacological treatments are available for AUD, our understanding of its mechanisms is limited [4,7,8,9]. Therefore, there is a pressing need to develop new and more effective pharmacological strategies to treat AUD. Preclinical models using mice, rats, and rhesus macaques have identified the neural and molecular substrates of alcohol consumption and probed the efficacy of novel molecules to mitigate the symptoms of AUD [10,11,12,13,14,15,16,17,18,19,20,21]. In vitro studies reported that through voltage-gated Ca^2+^ channels, Ca^2+^ influx contributes to regulating neuronal excitability and behavioral responses toward alcohol [22]. However, the role of Ca^2+^ entry through L-type of high threshold voltage-gated Ca^2+^ channels in alcohol consumption is complex, with conflicting reports. Some in vivo studies have reported that blockers of L-type Ca^2+^ channels reduced alcohol consumption [23,24]. Conversely, activators of L-type Ca^2+^ channels also reduced alcohol intake, and these effects were not antagonized by blockers of L-type Ca^2+^ channels [25]. Therefore, Ca^2+^ entry through other Ca^2+^ signaling mechanisms unrelated to the L-type Ca^2+^ channels may be responsible for reducing alcohol consumption. One potential Ca^2+^-dependent mechanism for alcohol consumption is Ca^2+^ entry through the sodium–calcium (Na^+^/Ca^2+^) exchanger (NCX), a bidirectional transporter that mediates an electrogenic exchange of three Na^+^ for one Ca^2+^ [26,27,28]. The NCX functions in two modes: the forward mode activity drives Ca^2+^ extrusion/Na^+^ entry coupling, while the reverse mode activity drives Ca^2+^ entry/Na^+^ extrusion coupling [26,27,28]. Three isoforms of NCX, including type 1 (NCX1), type 2 (NCX2), and type 3 (NCX3), have been identified with NCX1 and NCX3 having distinct pharmacological sensitivity [26,27,28]. Previous in vivo studies have reported that inhibiting NCX1 reverse mode activity can reduce the occurrence of acoustically evoked generalized tonic–clonic seizures (GTCSs) in a model of inherited epilepsy [29]. Interestingly, another in vivo study has also found that inhibiting the reverse mode activity of NCX1 and, to a lesser extent, NCX3 can suppress acoustically evoked GTCSs following alcohol withdrawal [30]. Furthermore, other studies have reported an increase in the levels of expression of NCX1 and NCX3 in the inferior colliculus (IC), which is the site for initiation of acoustically evoked seizure activity [31]. The targeted inhibition of NCX1 and NCX3 reverse mode activity within the IC was found to prevent the occurrence of alcohol withdrawal-induced GTCSs [31]. These findings demonstrate the potential of blocking NCX1 reverse mode activity in suppressing GTCSs of different etiologies and suggest that NCX reverse mode activity can play a role in controlling alcohol intake.

To test this hypothesis, we evaluated the effectiveness of SN-6 and KB-R7943, potent inhibitors of the reverse mode activity of NCX1 and NCX3, respectively, in modifying alcohol consumption and preference in male and female Sprague Dawley rats. The present study is unique because our previous studies focused solely on the effects of NCX reverse mode activity inhibitors on seizures in models of inherited epilepsy and alcohol withdrawal.

## 2. Results

Numerical data were analyzed using a general linear regression. The analysis revealed significant differences in population means for sex (F_6,1_ = 19.852, *p* = 0.00001), dose (F_6,1_ = 20.938, *p* = 0.00001), time (F_6,1_ = 138.114, *p* = 0.000001), and inhibitors (F_6,1_ = 72.983, *p* = 0.000001).

### 2.1. Overall Effects of SN-6 on Ethanol and Water Consumption

#### 2.1.1. Ethanol Intake

We first examined the effects of treatment (vehiclepretreatment, SN-6, and vehicleposttreatment), sex (male, female), SN-6 doses (3 and 10 mg/kg p.o.), and their interactions to ethanol intake 2 h after exposure to water and 7.5% ethanol (Table 1). Three-way ANOVA revealed significant differences for sex (*p* < 0.0001) and treatment (*p* = 0.01). Significant interactions were found between females and males for 3 mg/kg (*t* = 7.12, *p* < 0.001) and 10 mg/kg (*t* = 1.33, *p* = 0.002) SN-6 doses. We also analyzed the effects of treatment, sex, and SN-6 doses and their interactions on ethanol intake at the 24th hour after exposure to water and 7.5% ethanol (Table 1). Analysis revealed significant differences for sex (*p* < 0.0001) and treatment (*p* = 0.05). A significant interaction was found between sex and doses (*p* = 0.009). Moreover, significant interactions were observed between females and males for the 3 mg/kg (*t* = 8.56, *p* < 0.001) and the 10 mg/kg (*t* = 5.31, *p* = 0.001) SN-6 doses.

#### 2.1.2. Preference

Next, we examined the effects of different treatments (vehicle pretreatment, SN-6, and vehicle posttreatment), sex (male and female), and SN-6 doses (3 and 10 mg/kg p.o.) and their interactions toethanol preference at the 2nd hour after exposure to water and 7.5% ethanol (Table 1). Three-way ANOVA showed significant differences for sex (*p* = 0.002) and SN-6 doses (*p* < 0.001) but not for treatment. We also found significant interactions between SN-6 doses and treatment (*p* = 0.002), sex, SN-6 doses, and treatment (*p* = 0.05). Moreover, significant interactions were observed between males and females for 3 (*t* = 8.42, *p* < 0.001) and 10 mg/kg (*t* = 3.22, *p* < 0.001) SN-6 doses. We also analyzed the effects of treatment, sex, and SN-6 doses and their interactions 24 h after exposure to water and 7.5% ethanol (Table 1). Analysis showed significant differences for SN-6 doses (*p* < 0.001). There were also significant interactions between 3 and 10 mg/kg SN-6 doses in both males (*t* = 15.44, *p* < 0.001) and females (*t* = *p* < 0.001).

#### 2.1.3. Water

We examined the effects of treatment (vehicle pretreatment, SN-6, and vehicle posttreatment), sex (male and female), and SN-6 doses (3 and 10 mg/kg p.o.), as well as their interactions to water intake 2 h after exposure to water and 7.5% ethanol (Table 1). Three-way ANOVA revealed significant differences for sex (*p* = 0.034). Next, we analyzed the effects of treatment, sex, and SN-6 doses and their interactions in water intake at the 24th hour after exposure to water and 7.5% ethanol (Table 1). Analysis showed no statistical differences for sex, treatment, and SN-6 and their interactions.

### 2.2. Overall Effects of KB-R7934 on Ethanol and Water Consumption

#### 2.2.1. Ethanol Intake

We first analyzed the effects of treatment (vehicle pretreatment, KB-R7943, and vehicle posttreatment), sex (male and female), and KB-R7943 doses (3 and 10 mg/kg p.o.), as well as their interactions to ethanol intake at the 2nd hour after exposure to water and 7.5% ethanol (Table 2). Three-way ANOVA revealed significant differences for sex (*p* < 0.0001) but not for KB-R7943 doses or treatment. The analysis also showed significant interactions between female and male rats for 3 mg/kg (*t* = 4.89, *p* = 0.004) and 10 mg/kg (*t* = 5.97, *p* < 0.001) KB-R7943 dose. We also examined the effects of treatment, sex, KB-R7943 doses, and their interactions in ethanol intake at the 24th hour after exposure to water and 7.5% ethanol (Table 2). Analysis revealed significant differences for sex (*p* < 0.0001) and treatment (*p* = 0.01).

#### 2.2.2. Preference

Next, we examined the effects of treatment (vehicle pretreatment, SN-6, and vehicle posttreatment), sex (male andfemale), KB-R7943 doses (3 and 10 mg/kg p.o.), and their interactions to ethanol preference at the 2nd hour after exposure to water and 7.5% ethanol (Table 2). Three-way ANOVA revealed significant differences for sex (*p* = 0.01) and doses (*p* = 0.0002). Moreover, there was a significant interaction between sex and KB-R7943 doses (*p* = 0.01) but not between sex and treatment or sex and KB-R7943 doses. Furthermore, a significant interaction was observed between 3 and 10 mg/kg KB-R7943 doses in males (*t* = 2.71, *p* < 0.001) but not in females. Additionally, a significant interaction was observed between males and females during the 10 mg/kg KB-R7943 dose test (*t* = 3.88, *p* = 0.001). We also examined the effects of treatment, sex, and KB-R7943 doses and their interactions in ethanol preference 24 h after exposure to water and 7.5% ethanol (Table 2). Analysis revealed significant differences for sex (*p* = 0.031). A significant interaction was observed between sex and KB-R7943 doses (*p* = 0.03).

#### 2.2.3. Water

We analyzed the effects of treatment (vehicle pretreatment, KB-R7943, and vehicle posttreatment), sex (male and female), KB-R7943 doses (3 and 10 mg/kg p.o.), and their interactions to water intake at the 2nd hour after exposure to water and 7.5% ethanol) (Table 2). Three-way ANOVA revealed significant differences for sex (*p* = 0.002) and treatment (*p* = 0.02) but not KB-R7943 doses. Additionally, there were no statistical differences in interaction between sex and KB-R7943 doses, KB-R7943 doses and treatment, sex and treatment or sex, KB-R7943 doses, and treatment. We also examined the effects of treatment, sex, and KB-R7943 doses and their interactions to water intake 24 h after exposure to water and 7.5% ethanol (Table 1). Analysis revealed no statistical differences for sex, treatment, and dose factors and interactions between factors.

### 2.3. Effects of SN-6 Administration on Ethanol Intake and Preference

The effects of SN-6 treatment at a dose of 3 mg/kg (p.o.) on ethanol intake and preference were examined in males and females 2 and 24 h after exposure to water and 7.5% ethanol. One-way repeated measures (RM) ANOVA showed significant differences in ethanol intake among males (F_2,22_= 6.15, *p =* 0.007) at the 2nd hour time point but not in females (*p =* 0.48). Further analysis revealed a long-lasting decrease in ethanol intake after SN-6 treatment in males (vehicle pretreatment vs. 3mg/kg SN-6: *t =* 2.42, *p* = 0.007 and vehicle pretreatment vs. vehicle posttreatment: *t* = 3.40, *p =* 0.007, Figure 1, panel B). However, there were no statistical differences in the amount of ethanol intake by females (*p =* 0.14, Figure 1, panel A) and males (*p* = 0.06, Figure 1, panel B) at the 24th hour time point after exposure to water and 7.5% ethanol. We also analyzed the effects of SN-6 treatment at a dose of 10 mg/kg (p.o.). One-way RM ANOVA showed a significant decrease in ethanol intake 2 h after exposure to water and 7.5% ethanol in male rats (F_2,22_= 3.78, *p =* 0.03; Figure 1, panel D) but not in females (*p =* 0.37; Figure 1, panel C). Further analysis revealed that ethanol intake decreased compared to the vehicle pretreatment in male rats (*t =* 2.74, *p* = 0.03, Figure 1, panel D). Quantification also showed that SN-6 at a dose of 10 mg/kg (p.o.) significantly decreased ethanol intake at the 24th hour time point in female rats (F_2,22_ = 11.63, *p =* 0.0003, Figure 1, panel C). Further analysis revealed that this effect was long-lasting (vehicle pretreatment vs. 10 mg/kg SN-6: *t* = 4.44, *p =* 0.006; and vehicle pretreatment vs. vehicle posttreatment: *t* = 3.84, *p =* 0.002, Figure 1, panel C). In males, one-way RM ANOVA showed that ethanol intake was significantly reduced (F_2,22_ = 3.87, *p* = 0.03). Further analysis revealed that this effect was acute but not long-lasting (vehicle pretreatment vs. 10 mg/kg SN-6; *t* = 2.68, *p* = 0.004, Figure 1, panel D).

We also analyzed the effects of SN-6 (3 or 10 mg/k; p.o.) on ethanol preference measured 2 and 24 h after exposure to water and 7.5% ethanol. One-way RM ANOVA showed that 3 mg/kg (p.o.) SN-6 treatment did not alter ethanol preference in both male and female rats at both the 2nd hour (females: *p =* 0.67, Figure 1, panel A and males: *p* = 0.65, Figure 2, panel B) and the 24th hour (females: *p =* 0.97, Figure 2, panel A and males: *p* = 0.45, Figure 2, panel B) time points after exposure to water and 7.4% ethanol. However, the administration of SN-6 at the dose of 10 mg/kg (p.o.) significantly decreased ethanol preference in females (F_2,22_ = 5.42, *p* = 0.01, Figure 2, panel C) but not in males (*p* = 0.37, Figure 2, panel D) 24 h after exposure to water and 7.5% ethanol. Further analysis revealed that the inhibitory effects of SN-6 in ethanol preference in females (vehicle pretreatment vs. 10 mg/kg SN-6: *t =* 2.80, *p* = 0.03) were long-lasting (vehicle pretreatment vs. vehicle posttreatment: *t* = 2.90, *p =* 0.02, Figure 2, panel C). No statistical differences were found at the 2nd hour time point in males (*p* = 0.25; Figure 2, panel D) and females (*p* = 0.12, Figure 2, panel C).

### 2.4. Effects of Administration KB-R7943 on Ethanol Intake and Preference

We analyzed the effects of KB-R7943 (3 or 10 mg/kg, p.o.) treatment on ethanol intake and preference in males and females 2 and 24 h after exposure to water and 7.5% ethanol. One-way RM ANOVA showed that KB-R7943 administration at a dose of 3 mg/kg (p.o.) did not affect ethanol intake at the 2nd hour time point in both females (*p* = 0.45, Figure 3, panel A) and males (*p* = 0.10, Figure 3, panel B). However, 3 mg/kg (p.o.) KB-R7943 treatment significantly decreased ethanol intake at the 24th hour time point in females (F_2,22_= 3.93, *p* = 0.03, Figure 3, panel A) but not in males (*p* = 0.09, Figure 3, panel B). Further analysis revealed that KB-R7943 treatment resulted in lower ethanol intake than vehicle pretreatment (*t* = 2.80, *p* = 0.03, Figure 3, panel A). After administrating KB-R7943 at a dose of 10 mg/kg (p.o.), the results showed no statistical differences in ethanol intake 2 h after exposure to water and 7.5% ethanol in both females (*p*= 0.15, Figure 3, panel C) and males (*p* = 0.52, Figure 3, panel D). However, at the 24th hour time point, the analysis showed significant differences in ethanol intake in female rats (F_2,22_ = 6.15, *p* = 0.007, Figure 3, panel C) but not in males (*p* = 0.93, Figure 3D). Further analysis revealed that KB-R7943 treatment significantly decreased ethanol intake in female rats (vehiclepretreatment vs. 10 mg/kg KB-R7943; *t* = 3.48, *p* = 0.006, Figure 3, panel C), but this inhibitory effect was not long-lasting.

We also analyzed the effects of KB-R7943 (3 or 10 mg/kg; p.o.) on ethanol preference measured at the 2nd and 24th hours after exposure to water and 7.5% ethanol. One-way RM ANOVA showed no statistical differences in ethanol preference in females (*p* = 0.93, Figure 4, panel A) and males (F_2,22_ = 5.06, *p* = 0.015, Figure 4, panel B) at the 2nd hour time point with a dose of 3 mg/kg KB-R7943. No statistical differences were also observed in ethanol preference at the 24th hour time point in females (*p* = 0.91, Figure 4, panel A) and males (*p* = 0.32, Figure 4, panel B). After the administration of 10 mg/kg KB-R7943, no statistical differences in ethanol preference were found in either females or males 2 h after exposure to water and 7.5% ethanol (female: *p* = 0.60, panel C, and male: *p* = 0.65, Figure 4, panel D). There were also no statistical differences in ethanol preference found in females (*p* = 0.75, Figure 4, panel C) and males (*p* = 0.68, Figure 4, panel D) 24 h after exposure to water and 7.5% ethanol.

### 2.5. Effects of SN-6 or KB-R7943 on Water Intake

We analyzed the effects of two doses of SN-6 (3 or 10 mg/kg, p.o.) or KB-R7943 (3 or 10 mg/kg, p.o.) on water preference in male and female rats. Analysis showed that a 3 mg/kg dose of SN-6 did not affect water intake levels in females (*p* = 0.42; Figure 5, panel A) and males (*p* = 0.57, Figure 5, panel B) 2 h after exposure to water and 7.5% ethanol. Similarly, a 10 mg/kg dose of SN-6 did not cause any statistical difference in water intake levels in females (*p* = 0.16; Figure 5, panel C) and males (*p* = 0.76, Figure 5, panel D) 2 h after exposure to water and 7.5% ethanol. Additionally, there were no statistical differences in water intake levels 24 h after exposure to water and 7.5% ethanol under both 3 mg/kg SN-6 (female: *p* = 0.49, and male: *p* = 0.08, Figure 5, panels A,B) and 10 mg/kg SN-6 (male: *p* = 0.76, and female: *p* = 0.22, Figure 5, panels C,D). Quantification also showed that KB-R7943 administration at the dose of 3 mg/kg (p.o.) did not change water intake levels 2 h after exposure to water and 7.5% ethanol among both females (*p* = 0.07; Figure 5, panel E) and males (*p* = 0.06, Figure 5, panel F). Similarly, there were no statistical differences in water intake levels 2 h after water and 7.5% ethanol exposure under 10 mg/kg (p.o.) KB-R7943 in females (*p* = 0.88; Figure 5, panel G) and males (*p* = 0.77, Figure 5, panel H). Additionally, there were no statistical differences in water intake levels 24 h after water and 7.5% ethanol exposure under 3 mg/kg KB-R7943 in both females (*p* = 0.65; Figure 5, panel E) and males (*p* = 0.08, Figure 5, panel F) or 10 mg/kg KB-R7943 in both females (*p* = 0.93; Figure 5, panel G) and males (*p* = 0.77, Figure 5, panel H).

### 2.6. Evaluation of Proestrus and Estrus Stages

We also measured vaginal wall impedance in female rats during the test days. The mean vaginal impedance value was 0.51 ± 0.07 (range: 0.8 to 2.2 kΩ, n = 48), indicating the estrus stage.

## 3. Discussion

The outcomes of this study validate our working hypothesis that NCX reverse mode activity can impact alcohol consumption. Our main discovery is that the administration of SN-6, an NCX1 reverse mode activity inhibitor, significantly decreased alcohol intake and preference in male and female Sprague Dawley D rats. Other studies have also reported reduced alcohol consumption and preference in both males and females after using various pharmacological probes such as semaglutide, a glucagon-like peptide1 receptor agonist [11], desformylflustrabromine, a positive allosteric modulator of α4β2 nicotinic acetylcholine receptor [12], LY2817412, a selective nociception receptor blocker [15], sazetidine-A, an agonist of α4β2 and α7 nicotinic acetylcholine receptor [17], Apremilast, a phosphodiesterase type 4 inhibitor [19], and fenofibrate, a potent peroxisome proliferator-activated receptor agonist [20]. However, KB-R7943, an NCX3 reverse mode activity inhibitor, only reduced alcohol intake (but not alcohol preference) exclusively in females. These studies also suggest that there are diverse pharmacological mechanisms for controlling alcohol intake and preference. Our findings indicate that alcohol exposure may increase NCX reverse mode activity in non-alcohol-dependent male and female Sprague Dawley rats, consistent with an upregulation of NCX1 and NXC3 proteins in the IC during the earlier phase of alcohol withdrawal when blood alcohol levels are still high [31]. Further, blocking NCX1 operating in the reverse activity was effective in reducing alcohol consumption in both male and female rats at similar doses; this has potential therapeutic implications. Moreover, SN-6 and KB-R7943 did not affect water consumption, suggesting that NCX reverse mode inhibitors may not cause a general suppression of consummatory behavior, making them potential selective agents targeting alcohol intake. Interestingly, the selective reduction in alcohol consumption was associated with no notable side effects following oral administration [29,30,31,32,33]. Further, inhibiting NCX1 reverse mode activity (and, to some extent, NCX3 reverse mode activity) within the IC can also suppress alcohol withdrawal seizures in male rats following chronic alcohol intoxication [30]. These observations suggest that inhibitors of NCX reverse mode activity could be a viable option for AUD individuals who are more susceptible to alcohol withdrawal-induced seizures. Such an option may also be possible through a CB1 receptor antagonist AM6527 or by using sazetidine-A, an α4β2 and α7 nicotinic receptor agonist, both reduced handling-induced convulsion in mice following alcohol withdrawal [11,17].

Multiple lines of evidence suggest that sex-related factors can impact alcohol preference, which can be related to the molecular and neural basis of sex differences [34,35,36,37]. Sexual dimorphism within the brain reveals that females feature large caudate nuclei and cortices, whereas males feature larger amygdala [38,39]. These structural changes may contribute to sex differences in alcohol intake and preference. In this study, inhibiting NCX3 and NCX1 reverse mode activity selectively decreased alcohol intake and alcohol preference, respectively, in female but not in male rats. These findings suggest that there may be a differential expression of NCX3 and NCX1 in the brain networks of female and male rats in response to alcohol exposure. One brain network of interest is the mesocorticolimbic dopaminergic pathway, which is essential in motivation to seek alcohol and alcohol reward [13,21,40,41]. Evidence indicates that estrogens potentiate alcohol-induced excitation of neurons in the ventral tegmental area (VTA) [42]. Interestingly, the targeted knockdown of estrogen in the VTA reduces alcohol drinking in females but not in males [42,43]. Hence, we hypothesize that inhibiting NCX3 or NCX1 reverse mode activity may decrease activity in the mesolimbic dopaminergic pathway, which is responsible for estrogen-mediated facilitation of drinking in female rats. Additionally, female rodents typically consume more alcohol than males, which may be due to hyperactivity of the mesocorticolimbic dopaminergic pathway [36]. Our study also found that all female rats were in the estrus phase during the test days; this suggests that NCX reverse mode activity inhibitors may be more effective in reducing alcohol consumption during ovulation.

Evidence indicates that SN-6 and KB-R7943 have excellent brain permeability [29,30,32,33]. Although these compounds are potent inhibitors of the three NCX isotopes, they primarily target NCX1 (SN-6) and NCX3 (KB-R7943) reverse mode activity, respectively, with IC50 < 30 μM [32,33]. Therefore, it is probable that doses of 3 and 10 mg/kg could reduce alcohol consumption and preference when NCX is exclusively operating in the reverse mode in some brain sites.

The mechanisms underlying how inhibiting NCXrev activity reduces alcohol intake and preference are not fully understood. Multiple lines of evidence indicate that Ca^2+^ signaling is a target of the action of alcohol [22]. In particular, Ca^2+^ entry via L-type Ca^2+^ channels has been implicated in alcohol intake; however, the reports are conflicting [23,24,25]. These findings suggest that other Ca^2+^ entry mechanisms may play a critical role in alcohol consumption. One of the mechanisms is NCX operating in its reverse mode activity, as it can act as a Ca^2+^ entry route [26,27,28,29,30,31,32,33,34,35,36,37,38]. Interestingly, NCX is a target of the action of alcohol. Accordingly, reports indicated that chronic alcohol exposure causes a long-lasting increase in NCX protein expression in brain synaptic membranes and NCX activity in synaptosomes [44,45]. Furthermore, NCX1 and NCX3 protein levels were increased in the IC before the onset of alcohol withdrawal-induced seizure susceptibility [31]. Additionally, inhibiting NCX1rev (but not NCX3rev) within the IC decreased the occurrence of alcohol withdrawal-induced seizures and reduced their severity [31]. These data suggest that the IC may be a brain target where the inhibition of NCX reverse activity can suppress alcohol consumption. Interestingly, the IC can respond to rewarding and aversive stimuli [46,47,48]. NCX reverse mode activity inhibitors can reduce alcohol consumption by altering neuronal activity in the networks relevant to motivation to seek alcohol and alcohol reward [18]. Interestingly, NCX1 and NCX3 expression has been found in the VTA and the ventral striatum (or nucleus accumbens, NAc), respectively [49,50]. These brain sites are part of the mesolimbic dopaminergic pathway, which plays an essential role in impulsive behavior and alcohol addiction [51,52,53]. It is tempting to suggest that inhibiting NCX reverse mode activity can influence the activity of the mesolimbic dopaminergic system, resulting in reduced dopamine release in the NAc and, ultimately, reduced alcohol consumption. Another plausible hypothesis is that NCX reverse mode inhibitors can potentially help reduce alcohol consumption by promoting the transporter’s forward mode and reducing intracellular Ca^2+^ levels (Figure 6). This leads to the downregulation of Ca^2+^-dependent mechanisms, including small conductance Ca^2+^-activated K^+^ (SK) channels. However, the role of SK channels in alcohol consumption is complex, with conflicting reports. Some studies report that focal microinjections of 1-EBIO, a blocker of SK channels within the NAc, can decrease alcohol intake [54]. Other studies suggest that focal infusions of apamin, another blocker of SK channels within the NAc, increased alcohol consumption [55]. Therefore, it is unlikely that NCX reverse mode activity inhibitors reduce alcohol consumption via SK channel mechanisms. Other potential Ca^2+^ signaling molecular targets include the large conductance, Ca^2+^-activated K^+^ channels, Ca^2+^-activated chloride channels, and intracellular Ca^2+^ release channels. Additionally, inhibiting NCX reverse mode activity can raise intracellular Na^+^ levels, which may result in the upregulation of Na^+^-dependent mechanisms such as Na^+^-activated K^+^ channels. Future studies would be interesting in probing the role of Na^+^-activated K^+^ channels in regulating voluntary alcohol consumption and preference.

## 4. Conclusions

This report presents the first evidence that the administration of NCX reverse mode activity inhibitors can decrease alcohol intake and preference in rats and highlights the role of NCX1 and NCX3 in controlling alcohol consumption in males and females, and females only, respectively. Further studies are necessary to validate these findings in other species, such as mice and rhesus macaques. This study is significant as it demonstrates the potential of NCX reverse mode activity inhibitors to decrease alcohol consumption, which is consistent with the notion that alcohol exposure can increase NCX reverse mode activity. Notably, NCX reverse mode inhibitors can also suppress alcohol withdrawal seizures. Thus, inhibiting NCX activity may be a promising approach for treating AUD in individuals who are more susceptible to alcohol withdrawal seizures.

## 5. Materials and Methods

### 5.1. Animals

We used 8-week-old Sprague Dawley rats (48 males and 48 females, 270–300 g). These rats were obtained from Taconic Bioscience (Germantown, NY, USA). Animals were kept in a well-maintained room with controlled temperature and humidity and a 12 h light/dark cycle; they had free access to food and water. We made efforts to minimize the number of animals used in experiments and followed the guidelines outlined by the Guide for the Care and Use of Laboratory Animals [56]. All experimental procedures were approved by the Institutional Animal Care and Use Committee under Protocol MED 20-03 on 24 July 2023.

### 5.2. Ethanol Consumption Paradigm: Intermittent Alcohol Access Two-Bottle Choice Paradigm

Ethanol solutions (7.5% *v*/*v*) were prepared in purified water from ethanol 95% stock solution (U.S.P., The Warner-Gram Company, Cockeysville, MD, USA). The 7.5% ethanol dose was used based on our findings that SD rats consumed more 7.5% ethanol than 15 or 30% ethanol [57]. The fluids were presented in 40 mL graduated Drinkomeasurer bottles (Amuza, San Diego, CA, USA) with stainless steel sippers inserted through two eyelets at the front of the cage. We used an intermittent alcohol access two-bottle choice paradigm in which animals have access to ethanol without sweeteners during three sessions of 24 h per week [57,58,59]. After one week of acclimatization in their homecages, animals were housed individually and offered water from two Drinkomeasure bottles for acclimation for two weeks. All animals had simultaneous 24 h access to two bottles, with one containing ethanol 7.5% (*v*/*v*) and the other containing water, starting Monday at 10 a.m. Twenty-four hours later, the ethanol bottle was replaced with a second bottle of water that was available for the next 24 h. This pattern was repeated on Wednesdays and Fridays (Figure 7). On all other days, the rats had unlimited access to two water bottles. To prevent any side preferences, the placement of the bottles was alternated during each session. The bottles were weighed 2 and 24 h after fluid presentation, and the amount of ethanol and water was measured. Additionally, the body weight of the animals was measured before each session and monitored at 24 and 48 h during the session.

### 5.3. Drugs and Solutions

We used SN-6 (2-[[4-[(4-Nitrophenyl)methoxy]phenyl]methyl]-4-thiazolidine carboxylic acid ethyl ester) and KB-R7943 (2-[2-[4-(4-Nitrobenzyloxy)phenyl]ethyl]isothiourea mesylate) potent inhibitors of the NCX reverse mode activity for NCX1 and NCX3, respectively, which were purchased from R&D System (Minneapolis, MN, USA). SN-6 and KB-R7943 were dissolved in dimethyl sulfonic acid (0.1%) and phosphate-buffered saline (pH 7.4) using sonication (80 kHz, 100% power). The solutions containing either SN-6 (0, 3, 10 mg/kg, p.o.) or KB-R7943 (0, 3, 10 mg/kg, p.o.) were freshly prepared before their administration, as previously described [29,30]. After four weeks of training, animals were randomly assigned into eight groups (n= 12) including female KB-R7943-3 (KB-R7943, 3 mg/kg), female KB-R7943-10 (KB-R7943, 10 mg/kg), male KB-R7943-3 (KB-R7943, 3 mg/kg), male KB-R7943-10 (KB-R7943, 10 mg/kg), female SN-6-3 (KB-R7943, 3 mg/kg), female SN-6-10 (SN-6, 10 mg/kg), male SN-6-3 (SN-6, 3 mg/kg), and male SN-6-10 (SN-6, 10 mg/kg). On Mondays, animals received the vehicle; on Wednesdays, animals received the tested drug, either SN-6 or KB-R7943; and on Fridays, they were examined to see the long-lasting effects of the drugs.

### 5.4. Vaginal Impedance

The Rat Vaginal Impedance Checker (Model MK-11; Muromachi Kikai, Tokyo, Japan) was used to measure vaginal impedance. Measurements were taken daily between 1 and 3 pm. A plastic probe (4.5 mm in diameter) containing two silver ring electrodes was inserted into the vagina and held stable to ensure complete contact with the metal electrodes. A threshold ≥3.0 kΩ was used to confirm the proestrus phase, while the estrus phase was characterized by a <3.0 kΩ [60]. The vaginal impedance readings provide a more accurate method of determining phases of the estrous cycle than vaginal smear cytology [60,61].

### 5.5. Data Analysis

The investigators were blinded to group allocation during experiments, and Origin 2023b. 10.05.157 software (OriginLab, Northampton, MA, USA) was used for statistical analyses and to create graphs. The body weight of each rat was used to calculate the grams of ethanol intake per kilogram of body weight and milliliters of water intake per kilogram. Ethanol preference was calculated as the ratio of ethanol to total fluid intake at the 24th hour time point. Measurements (ethanol and water intake) were carried out on Tuesdays, Thursdays, and Saturdays and were used as averages to calculate alcohol intake, ethanol preference, and water intake. For ethanol consumption, dependent measures included ethanol dose, alcohol preference, and water intake. Numerical data were first analyzed using general linear regression. Before performing ANOVA, data were subjected to the Kolmogorov–Smirnov test for normality and Levene’s test for homogeneity of variances. The ethanol intake, ethanol preference, and water intake were first analyzed using three-way ANOVA with sex (male and female), dose (3, 10 mg/kg), and treatment (vehicle pretreatment, NCX inhibitor, and vehicle posttreatment) as a between-subject factor and the 2nd or 24th hour time points as within-subject factor. For each time point (2nd or 24th hour), dose (3 or 10 mg/kg), and sex (male or female), data were then analyzed using repeated measures of one-way ANOVA. Post hoc comparisons were performed with Bonferroni correction, which compares the difference between each pair of means with an appropriate adjustment for multiple testing (*t*). The cut-off for statistical significance was set at *p* < 0.05. Data are presented as percentages (%) for ethanol preference and mean ± S.E.M. for ethanol intake, water intake, and vaginal impedance.

## Figures and Tables

**Figure 1 ijms-25-04132-f001:**
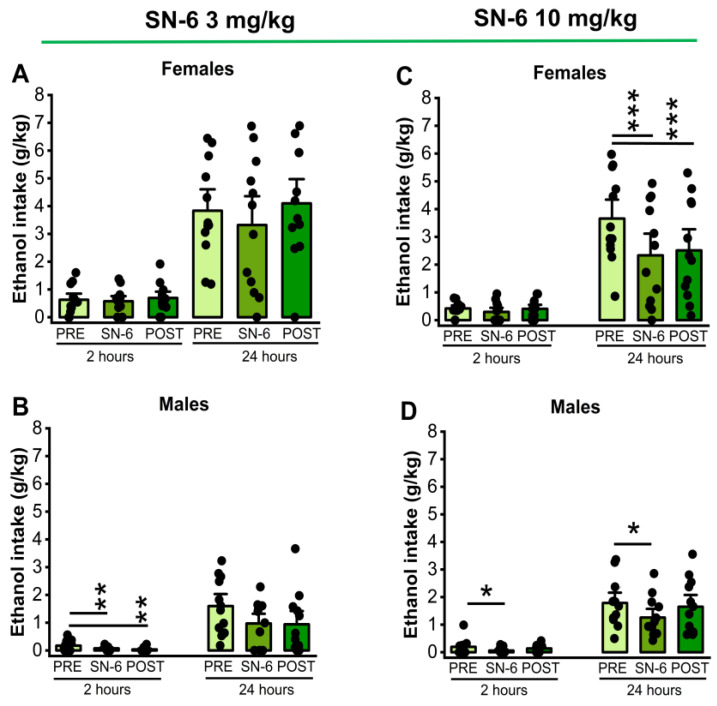
SN-6 and ethanol intake. SN-6 (3 or 10 mg/kg) was orally administered 30 min before exposure to water and 7.5% ethanol. At a dose of 3 mg/kg (p.o.), SN-6 significantly reduced ethanol intake in male (panel (**B**)) but not female (panel (**A**)) rats 2 h after exposure to water and 7.5% ethanol. Additionally, at a dose of 10 mg/kg (p.o.), SN-6 significantly reduced ethanol intake in both male and female rats; notably, the effect was more prominent in females and lasted longer (panels (**C**,**D**)). The filled dots represent individual data points. The data are presented as mean ± S.E.M for alcohol intake, and the analysis was performed using one-way repeated measures ANOVA followed by Bonferroni post hoc correction (* *p* < 0.05, ** *p* < 0.01 and *** *p* < 0.001, n = 12).

**Figure 2 ijms-25-04132-f002:**
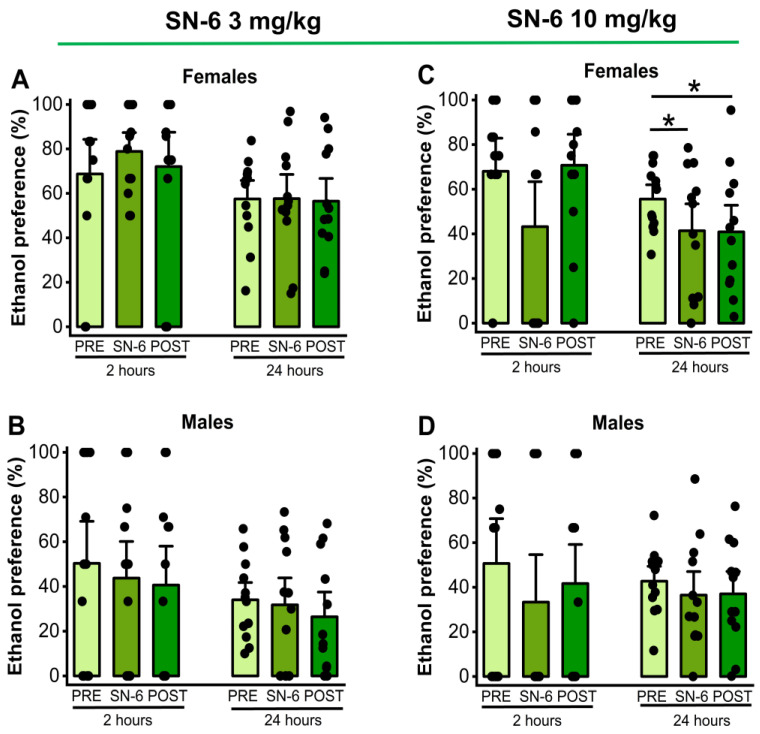
SN-6 and ethanol preference. SN-6 (3 or 10 mg/kg) was orally administered 30 min before exposure to water and 7.5% ethanol. When administered at 3 mg/kg (p.o.), SN-6 did not affect ethanol preference in either male or female rats (panels (**A**,**B**)). However, SN-6 at 10 mg/kg (p.o.) significantly decreased ethanol preference in female rats but not in male rats 24 h after exposure to water and 7.5% ethanol (panels (**C**,**D**)). The filled dots represent individual data points. The data are presented as percentages (%) and were analyzed using one-way repeated measures ANOVA followed by Bonferroni post hoc correction (* *p* <0.05, n = 12).

**Figure 3 ijms-25-04132-f003:**
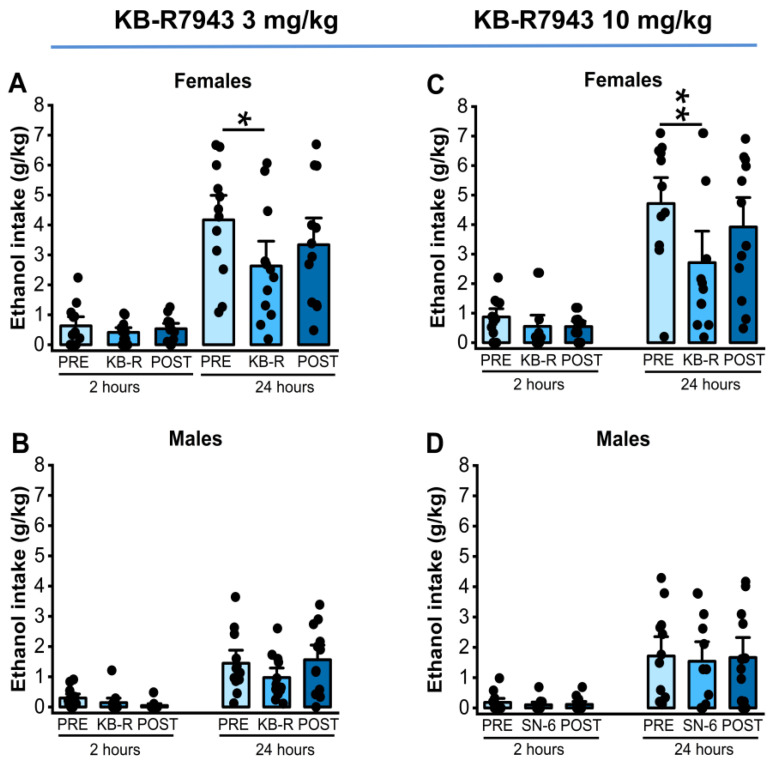
KB-R7943 and ethanol intake. KB-R7943 (3 or 10 mg/kg) was orally administered 30 min before exposure to water and 7.5% ethanol. KB-R7943 at the dose of 3 mg/kg (p.o.) treatment significantly decreased ethanol intake in female but not in male rats (**A**,**B**). Similarly, KB-R7943 at the dose of 10 mg/kg (p.o.) significantly decreased ethanol intake in female rats but not males (**C**,**D**). The filled dots represent individual data points. The data are presented as mean ± S.E.M and were analyzed using one-way repeated measures ANOVA followed by Bonferroni post hoc correction (* *p* < 0.05, ** *p* < 0.01, n = 12).

**Figure 4 ijms-25-04132-f004:**
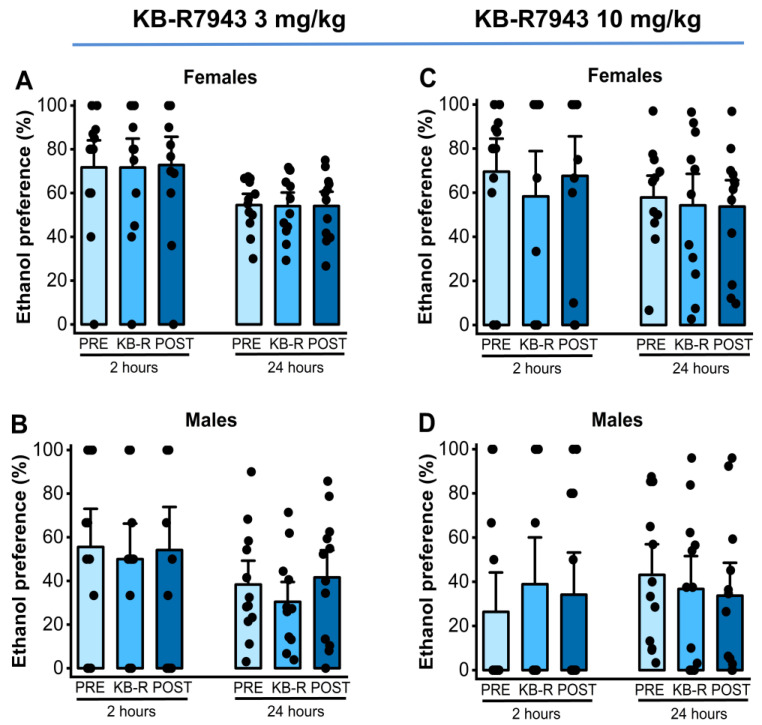
KB-R7943 and ethanol preference. KB-R7943 (3 or 10 mg/kg) was orally administered 30 min before exposure to water and 7.5% ethanol. When administered at a dose of 3 mg/kg (p.o.), KB-R7943 significantly decreased ethanol intake in males (panel (**B**)) but not in female rats (panel (**A**)). KB-R7943 at a dosage of 10 mg/kg (p.o.) did alter ethanol preference in female and male rats (panels (**C**,**D**)). The filled dots represent individual data points. The data are presented as percentages (%) and were analyzed using one-way repeated measures ANOVA followed by Bonferroni post hoc correction.

**Figure 5 ijms-25-04132-f005:**
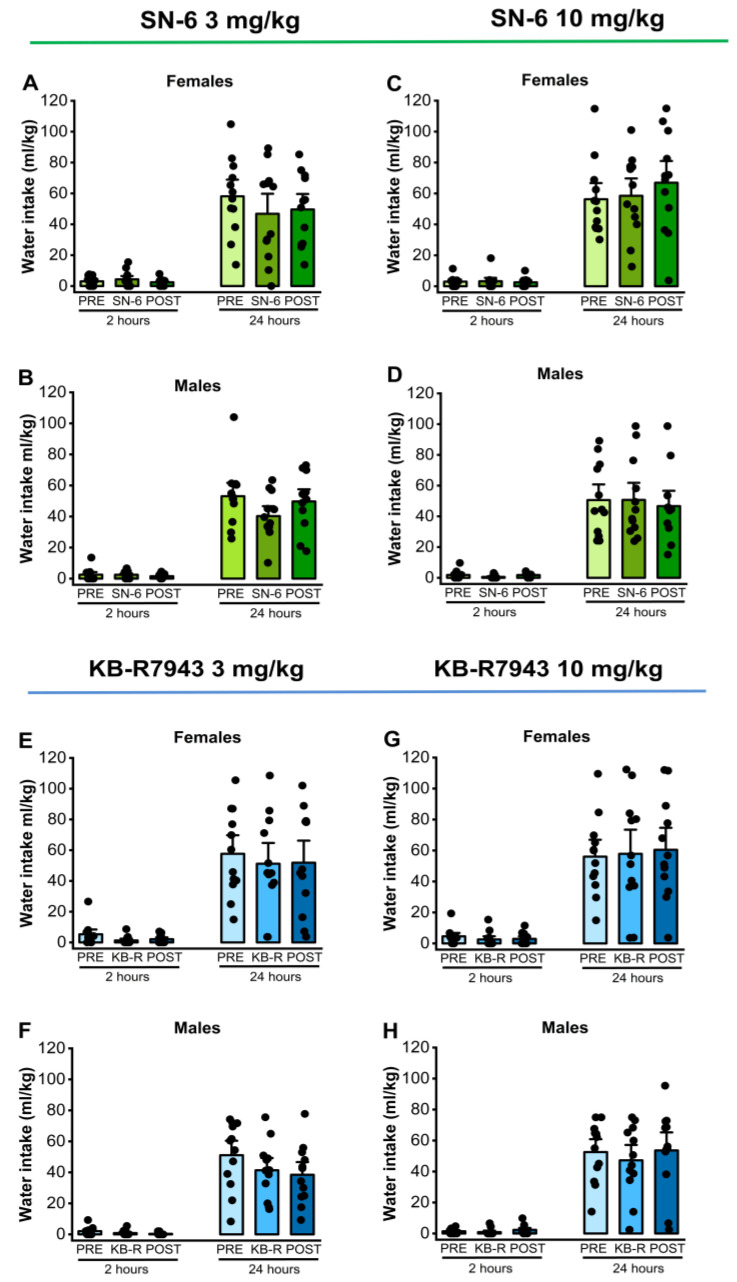
NCX inhibitors and water intake. SN-6 or KB-R7843 (3 or 10 mg/kg, p.o.) was orally administered 30 min before exposure to water and 7.5% ethanol. There were no considerable effects on water intake in female and male rats after SN-6 treatment (panels (**A**–**D**)). Similarly, the KB-R7943 treatment did not alter water intake in female and male rats (panels (**E**–**H**)). The filled dots represent individual data points. The data are presented as mean ± S.E.M for water intake and analyzed using one-way repeated measures ANOVA followed by Bonferroni post hoc correction (n = 12).

**Figure 6 ijms-25-04132-f006:**
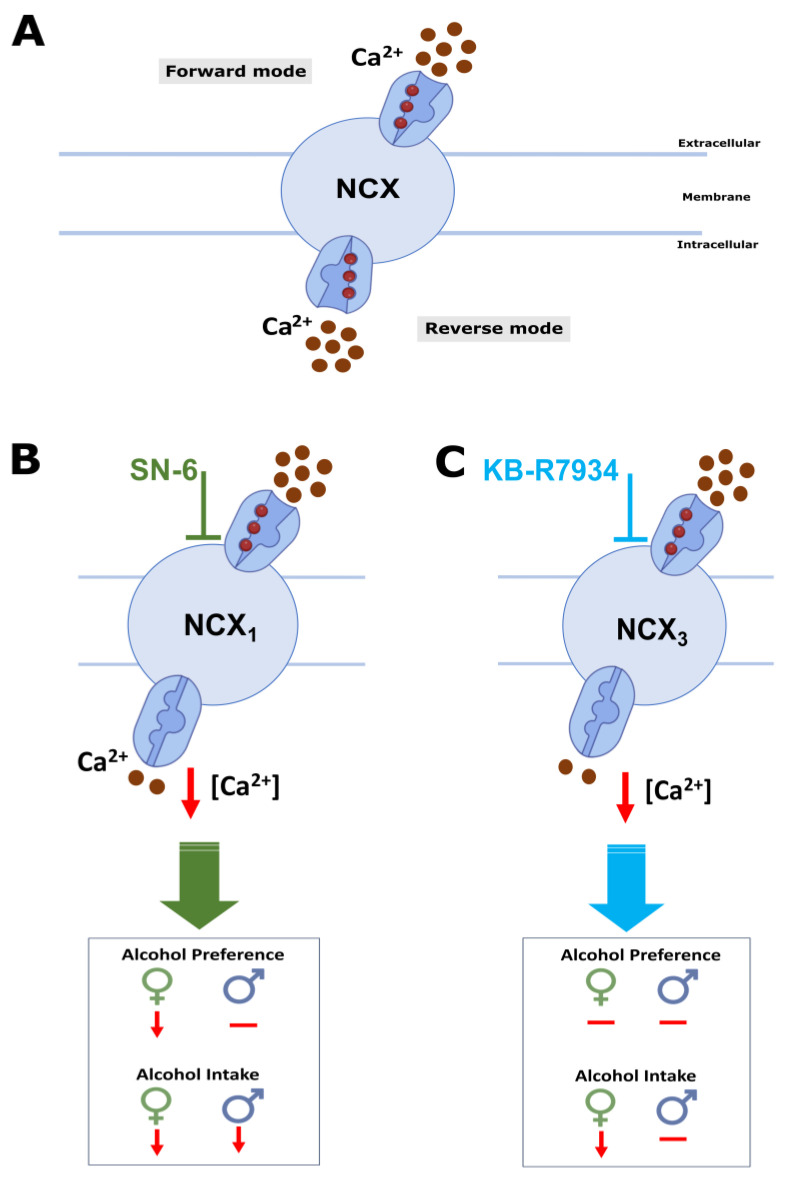
Summary and putative mechanisms. NCX activity results in Ca^2+^ influx in reverse mode and Ca^2+^ efflux during forward mode (**A**). Administration of NCX reverse mode activity inhibitors (**B**,**C**) decreases intracellular [Ca^2+^] levels, reducing Ca^2+^-dependent mechanisms (and activating Na^+^-dependent mechanisms), potentially leading to decreased ethanol consumption and preference.

**Figure 7 ijms-25-04132-f007:**
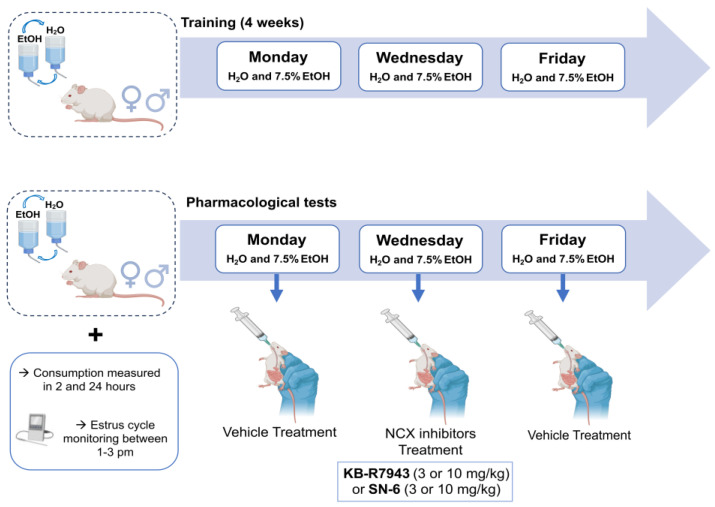
Experimental design. During the experiments, adult male and female rats were given a choice between water and 7.5% ethanol thrice a week (Mondays, Wednesdays, and Fridays) using a two-bottle choice paradigm. After four weeks of training, rats were given the vehicle (control solution) on Mondays, SN-6 or KB-R7943 on Wednesdays, and the vehicle (washout solution) on Fridays.

**Table 1 ijms-25-04132-t001:** Overall effects of SN-6 on ethanol intake, ethanol preference, and water consumption.

SN-6				
**Ethanol intake**	2 h		24 h	
Factor sex	F_1,132_ = 46.50	*p* < 0.0001	F_1,132_ = 59.4	*p* < 0.0001
Factor treatment	F_2,132_ = 4.34	*p* = 0.01	F_1,132_ = 3.009	*p* = 0.05
Factor SN-6 doses	F_1,132_ = 1.63	*p* = 0.20	F_1,132_ = 1.09	*p* = 0.29
Interaction sex and SN-6	F_1,132_ = 3.80	*p* = 0.053	F _1,132_ = 6.88	*p* = 0.009
Interaction SN-6 and treatment	F_2,132_ = 0.030	*p* = 0.73	F_2,132_ = 0.297	*p* = 0.74
Interaction sex and treatment	F_2,132_ = 1.82	*p* = 0.16	F_2,132_ = 0.18	*p* = 0.83
Interaction sex, SN-6 doses, and treatment	F_2,132_ = 1.32	*p* = 0.26	F_2,132_ = 1.24	*p* = 0.29
**Preference**	2 h		24 h	
Factor sex	F_1,132_ = 9.54	*p* = 0.002	F_1,132_ = 3.29	*p* = 0.07
Factor treatment	F_2,132_ = 0.04	*p* = 0.95	F_2,132_ = 1.71	*p* = 0.184
Factor SN-6 doses	F_1,132_= 65.08	*p* < 0.001	F _1,132_ = 218.6	*p* < 0.001
Interaction sex and SN-6	F_1,132_ = 0.13	*p* = 0.71	F_1,132_ = 0.73	*p* = 0.39
Interaction SN-6 and treatment	F_2,132_ = 6.29	*p* = 0.002	F_2,132_ = 1.47	*p* = 0.23
Interaction sex and treatment	F_2,132_ = 0.90	*p* = 0.40	F_2,132_ = 0.24	*p* = 0.78
Interaction sex, SN-6, and treatment	F_2,132_ = 3.03	*p* = 0.05	F_2,132_ = 0.31	*p* = 0.72
**Water**	2 h		24 h	
Factor sex	F_1,132_ = 4.55	*p* = 0.034	F_1,132_ = 3.48	*p* = 0.06
Factor treatment	F_2,132_ = 0.31	*p* = 0.73	F_2,132_ = 0.66	*p* = 0.51
Factor SN-6 doses	F_1,132_= 0.50	*p* = 0.47	F_1,132_= 1.74	*p* = 0.18
Interaction sex and SN-6	F_1,132_ = 0.54	*p* = 0.45	F_1,132_ = 0.84	*p* = 0.30
Interaction SN-6 and treatment	F_2,132_ = 0.41	*p* = 0.66	F_2,132_ = 0.94	*p* = 0.39
Interaction sex and treatment	F_2,132_ = 0.33	*p* = 0.71	F _2,132_ = 0.11	*p* = 0.88
Interaction sex, SN-6, and treatment	F_2,132_ = 0.33	*p* = 0.71	F_2,132_ = 0.64	*p* = 0.52

Three-way ANOVA was used to analyze the impact of sex (male and female), dose (3, 10 mg/kg), and treatment (vehicle pretreatment, NCX inhibitor, and vehicle posttreatment) on ethanol intake, ethanol preference, and water intake. The cut-off for statistical significance was set at *p* < 0.05 (n = 12).

**Table 2 ijms-25-04132-t002:** Overall effects of KB-R7934 on ethanol intake, ethanol preference, and water consumption.

KB-R7943				
**Ethanol intake**	2 h		24 h	
Factor sex	F_1,132_ = 32.77	*p* < 0.0001	F_1,132_ = 52.11	*p* < 0.0001
Factor treatment	F_2,132_ = 2.642	*p* = 0.07	F_2,132_ = 4.43	*p* = 0.01
Factor KB-R7943 doses	F_1,132_= 0.429	*p* = 0.513	F_1,132_= 1.50	*p* = 0.22
Interaction sex and KB-R7943	F_1,132_ = 1.085	*p* = 0.299	F_1,132_ = 0.002	*p* = 0.87
Interaction KB-R7943 and treatment	F_2,132_ = 0.008	*p* = 0.99	F_2,132_ = 0.007	*p* = 0.99
Interaction sex and treatment	F_2,132_ = 0.33	*p* = 0.71	F_2,132_ = 2.08	*p* = 0.12
Interaction sex, KB-R7943, and treatment	F_2,132_ = 0.59	*p* = 0.553	F_2,132_ = 0.25	*p* = 0.77
**Preference**	2 h		24 h	
Factor sex	F_1,132_ = 6.74	*p* = 0.01	F_1,132_ = 4.74	*p* = 0.031
Factor treatment	F_2,132_ = 0.04	*p* = 0.96	F_2,132_ = 0.29	*p* = 0.74
Factor KB-R7943 doses	F_1,132_ = 13.80	*p* = 0.0002	F_1,132_ = 3.66	*p* = 0.057
Interaction sex and KB-R7943	F_1,132_ = 6.74	*p* = 0.01	F_1,132_ = 4.74	*p* = 0.03
Interaction KB-R7943 and treatment	F_2,132_ = 0.007	*p* = 0.99	F_2,132_ = 0.22	*p* = 0.79
Interaction sex and treatment	F_2,132_ = 0.31	*p* = 0.73	F_2,132_ = 0.03	*p* = 0.96
Interaction sex, SN-6, and treatment	F_2,132_ = 0.31	*p* = 0.73	F_2,132_ = 0.036	*p* = 0.96
**Water**	2 h		24 h	
Factor sex	F_1,132_ = 9.17	*p* = 0.002	F_1,132_ = 3.57	*p* = 0.06
Factor treatment	F_2,132_ = 3.80	*p* = 0.02	F_2,132_ = 0.41	*p* = 0.66
Factor KB-R7943 doses	F_1,132_ = 0.63	*p* = 0.42	F_1,132_= 1.81	*p* = 0.18
Interaction sex and KB-R7943	F_1,132_ = 0.00009	*p* = 0.99	F_1,132_ = 0.109	*p* = 0.74
Interaction KB-R7943 and treatment	F_2,132_ = 1.06	*p* = 0.34	F _2,132_ = 0.59	*p* = 0.55
Interaction sex and treatment	F_2,132_ = 1.37	*p* = 0.25	F_2,132_ = 0.14	*p* = 0.86
Interaction sex, SN-6, and treatment	F_2,132_ = 0.32	*p* = 0.72	F_2,132_ = 0.056	*p* = 0.94

Three-way ANOVA was used to analyze the impact of sex (male and female), dose (3, 10 mg/kg), and treatment (vehicle pretreatment, NCX inhibitor, and vehicle posttreatment) on ethanol intake, ethanol preference, and water intake. The cut-off for statistical significance was set at *p* < 0.05 (n = 12)

## Data Availability

The data presented in this study are available upon request from the corresponding author.

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
