# Peer review of "Inhibition of the Sodium–Calcium Exchanger Reverse Mode Activity Reduces Alcohol Consumption in Rats"

_ijms, 2024, doi:10.3390/ijms25074132_

Round 1

Reviewer 1 Report

Comments and Suggestions for Authors

The authors describe the effect of NCX inhibitors on alcohol consumption in rats.

The following revisions might be considered:

Line 12: the pharmacology is limited sound strange. Our knowledge about it is perhaps limited. On the other hand, this is one of the best researched topics.

Line 24-24: the last sentence of the abstract is rather strange. The actual conclusion is that NCX inhibitors could be used for AUD patients, but you do not want a lifestyle drug for everyday use?

Line 30: avoid stigmatizing terms such as “abuse”

Line 44: please state for all studies in the introduction, if they were in vitro, or in vivo in animals or humans.

Line 51: superscript 2+ and +. Check throughout.

Results: the results section only contains textual values in very similar fashion for all the conditions. This is very tiresome to read. Consider adding some tables and mention only the “highlights” in the text.

Line 391: please shortly discuss what would be the next steps to make this applicable  for treatment of AUD.

Line 427: check chemical name

Author Response

Review 1

Point 1: Line 12: the pharmacology is limited sound strange. Our knowledge about it is perhaps limited. On the other hand, this is one of the best researched topics.

RE: We have rewritten the sentence: “Although pharmacological treatments are available for AUD, our understanding of its mechanisms is limited.” Lines 35-36

Point 2: The last sentence of the abstract is rather strange. The actual conclusion is that inhibitors could be used for AUD patients, but you do not want a lifestyle drug for everyday use?

RE: The last sentence has been rewritten as follows: “These findings suggest that alcohol exposure increased NCX reverse activity, and targeting NCX1 could be an effective strategy for reducing alcohol consumption in subjects susceptible to withdrawal seizures.” Line 23-25.

Point 3: Avoid stigmatizing terms such as “abuse.”

RE: We have removed this term from the manuscript. Line 30.

Point 4: Please state for all studies in the introduction, if they were in vitro, or in vivo in animals or humans.

RE: We agreed and made the necessary changes. Lines 40, 44, 55.

Point 5: Line 51: superscript 2+ and +. Check throughout.

RE: Thank you for your comment. We have made changes throughout the manuscript.

Point 6: Results: the results section only contains textual values in very similar fashion for all the condition. This is very tiresome to read. Consider adding some table and mention only the “highlights” in the text.

RE: Thank you for the suggestion. We have added Table 1 (page 3) and Table 2 (page 5) whenever applicable.

Point 7: Results: Line 391: please shortly discuss what would be the next steps to make this applicable for treatment of AUD.

RE: We have briefly discussed the next steps in lines 402-405, 419, 420.

Point 8: Line 427: check chemical name.

RE: Thank you for your feedback; we have checked the chemical names: SN-6 (2-[[4-[(4-Nitrophenyl)methoxy]phenyl]methyl]-4-thiazolidinecarboxylic acid ethyl ester) and KB-R7943 (2-[2-[4-(4-Nitrobenzyloxy)phenyl]ethyl]isothiourea mesylate).

Reviewer 2 Report

Comments and Suggestions for Authors

The authors aimed to evaluate the efficacy of two sodium-calcium exchanger (NCX) reverse-mode inhibitors [SN-6 (NCX1) and KB-R7943 (NCX3); two doses: 0, 3 and 10 mg.kg-1] in modifying alcohol (EtOH 7.5% w.w-1) and water intake & preference (2 and 24h post exposure; intermittent alcohol access two-bottle choice paradigm) by male (48):female (48) Sprague-Dawley rats. The deviations of each response variable (EtOH or water intake/preference) were recorded by time/dose/sex/inhibitor (Figures 2-6). Authors were able to demonstrate that SN-6 (sex independent) and KB-R7943 (sex dependent) significantly reduced alcohol intake more than preference. The experimental design/execution are impeccable, and the description/discussion of results based on evidence. However, some changes are necessary to improve the scientific soundness and uniqueness of the study.

General. B) Do not forget to describe the meaning of each abbreviation the first time it is mentioned.

Abstract. A) It should be described in a more quantitative than narrative way.

Introduction. A) Authors must highlight the uniqueness of this new study in relation to their previous ones (references 17, 22, 35, 38).

Methods. A) The authors should consider using multinomial logistic regression (or some other statistical tool) to estimate the participation of each independent variable (sex, dose, time, inhibitor) in the total variance associated to each response variable.

Results. A) Although the effects of confounding variables (sex, dose, type of inhibitor, etc.) They are easily seen in the graphs, the use of logistic regression to demonstrate the strength of association (% of the variance) of each of them with the response variables, would improve the understanding of the phenomena involved.

Discussion. A) The authors should delve even further into the biological determinants associated with sex and/or drive/preference for alcohol, possibly proposing one or two hypotheses supported by some graphic summary.

Figures. A) All should be provided with a higher resolution (>300 dpi). B) A figure depicting all plausible differential inhibitory mechanisms (summary of all the results) would be very useful. C) Figure titles must be differentiated from their corresponding footnotes. D) Figure 1 must be located in methods.

References. It is recommended to reduce old references (≥10 y) to say 25% or less (currently 62%).

Author Response

Review 2

Point 1: General. B) Do not forget to describe the meaning of each abbreviation the first time it is mentioned.

RE: Thank you for the suggestion. We have made the necessary changes where appropriate.

Point 2: Abstract. A) It should be more quantitative than narrative way.

RE: We have made the necessary changes based on the feedback. Lines 20-22.

Point 3: Introduction. A) The authors must highlight the uniqueness of this new study in relation to their previous ones (references 17, 22, 35, 38).

RE: We have considered your suggestions and made the necessary changes. Lines 55-65.

Point 4: Methods A) The authors should consider using multinominal logistic regression (or some other statistical tool) to estimate the participation of each independent variable (sex, dose, time, inhibitor) in total variance associated to each response variable.

RE: Thank you for your suggestion. Our numeric data have been analyzed using general linear regression. Lines 75-77, 502-503.

Point 5: Results A) Although the effects of confounding variables (sex, dose, type of inhibitor, etc.) They are easily seen in the graphs, the use of logistic regression to demonstrate the strength of the association (% of the variance) of each of them with the response variables, would improve the understanding of the phenomena involved.

RE: Please see Point 4.

Point 6: Discussion A) The authors should delve even further into the biological determinant associated with sex and/or drive/preference for alcohol, possibly proposing one or two hypotheses supported by some graphical summary.

RE: Thank you for your valuable input. We have the necessary revisions to the manuscript. Lines 342-359.

Point 7: Figures

  1. All should be provided with a high resolution (>300 dpi)
  2. A figure depicting all possible differential inhibitory mechanisms (summary of all results) would be very useful.
  3. Figure titles must be differentiated from their corresponding footnotes.
  4. Figure 1 must be located in methods.

RE:

  1. The figures are now in high resolution (600 dpi).
  2. A figure depicting a summary of all results is provided (Figure 6, page 13).
  3. All figure titles are now clearly distinguished from their corresponding footnotes. Lines 201, 223, 250, 269, and 302.
  4. Figure 1 has been relocated to Methods (page 1, Figure 7).

Point 8: References. It is recommended to reduce old references (>10 y) to say 25% or less (currently 62%).

RE: Thank you for your comment; we made changes accordingly when appropriate. About 23% of the references listed are older than 10 years.

Reviewer 3 Report

Comments and Suggestions for Authors

The article addresses a topic of extreme interest. Basically, the article provides a description of the results obtained with little comparison to the literature or proposed molecular mechanisms. Despite this, the article is well-written; from my perspective, it could be published after minor revisions.

-Given the importance of the subject matter, I expected more recent studies to be included in the introduction. The authors cited only 1 from 2022, 1 from 2024, and 1 from 2021 among the current ones, with the rest being older articles in the literature. The authors should conduct a slightly more detailed review and expand the introduction by adding more recent studies.

-The authors did not conclude their work; they provided a brief conclusion in the discussions. The authors should include a conclusion section.

-The authors obtained better results for females. Is there any evidence as to why? It would be interesting to include this discussion in the article.

-In the discussions, the authors merely describe the results obtained without comparing them to the literature. It would be beneficial to compare with results from the literature, especially with other animal models if available.

-The molecular interaction mechanisms were not proposed. The authors should add these mechanisms to the article by incorporating more data from the literature and proposing hypotheses.

Comments on the Quality of English Language

Minor editing of English language required

Author Response

Review 3

Point 1: Given the importance of the subject matter, I expected more recent studies to be included in the introduction. The authors cited only 1 from 2022, 1 from 2024, and 1 from 2021 among the current ones, with the rest being older literature. The authors should conduct a slightly more detailed review and expand the introduction by adding more recent studies. 

RE: Thank you for the suggestion; we have added 27 new references.

Point 2: The authors did not conclude their work; they provided a brief conclusion in the discussions. The authors should include a conclusion section. 

RE: We have included a Conclusion section.

Point 3: The authors obtained better results for females. Is there any evidence as to why? It would be interesting to include this discussion in the article. 

RE: We have carefully considered your suggestions and made the necessary changes. Lines 342-359.

Point 4: In the discussions, the authors merely describe the results obtained without comparing them to the literature. It would be beneficial to compare with results from the literature, especially with other animal models if available.  

RE: Thank you for your suggestion. We have compared and discussed our data with the literature when possible. Lines 312-324, 338-341, and 383-398.

Point 5: The molecular interaction mechanisms were not proposed. The authors should add these mechanisms to the article by incorporating more data from the literature and proposing hypothesis.

RE: We have made the necessary changes based on your suggestion.  Lines 391-405 and Figure 6 (page 13).

Round 2

Reviewer 2 Report

Comments and Suggestions for Authors

Thank you for accepting most, if not all, of my suggestions. The manuscript improved substantially.

Comments on the Quality of English Language

OK